# Association between Conflicts of Interest Disclosure and Quality of Clinical Practice Guidelines in Japan: A Meta-Epidemiological Study

**DOI:** 10.3390/jpm13121722

**Published:** 2023-12-17

**Authors:** Norio Yamamoto, Akihiko Ozaki, Shunsuke Taito, Takashi Ariie, Hidehiro Someko, Hiroaki Saito, Tetsuya Tanimoto, Yuki Kataoka

**Affiliations:** 1Department of Epidemiology, Graduate School of Medicine, Dentistry and Pharmaceutical Sciences, Okayama University, Okayama 700-8558, Japan; 2Scientific Research WorkS Peer Support Group (SRWS-PSG), Osaka 541-0043, Japan; 3Medical Governance Research Institute, Tokyo 113-8510, Japan; 4Department of Breast and Thyroid Surgery, Jyoban Hospital of Tokiwa Foundation, Fukushima 972-8322, Japan; 5Division of Rehabilitation, Department of Clinical Practice and Support, Hiroshima University Hospital, Hiroshima 734-0037, Japan; 6Department of Physical Therapy, School of Health Sciences at Fukuoka, International University of Health and Welfare, Fukuoka 286-8686, Japan; 7Department of General Internal Medicine, Asahi General Hospital, Chiba 289-2511, Japan; 8Department of Internal Medicine, Soma Central Hospital, Fukushima 975-0033, Japan; 9Department of Internal Medicine, Navitas Clinic, Tokyo 160-0022, Japan; 10Department of Internal Medicine, Kyoto Min-Iren Asukai Hospital, Kyoto 616-8147, Japan; 11Department of Healthcare Epidemiology, Kyoto University Graduate School of Medicine/School of Public Health, Kyoto 606-8501, Japan

**Keywords:** AGREE II, clinical guidelines, conflicts of interest, funding, guideline committee, guideline development process, Japan

## Abstract

Accurate disclosure of financial conflicts of interest (COI) among clinical practice guideline (CPG) developers is critical to ensure the quality of CPGs. However, there is limited evidence on the impact of underreporting COIs on the quality of CPGs. This study aimed to examine the proportion of underreported COI disclosures in the development of Japanese CPGs and to estimate the association between underreported COIs and CPG quality using the Appraisal of Guidelines for Research & Evaluation (AGREE) II. Twenty-three Japanese CPGs published in 2019 and their 1114 developers were included in the study. The results show that underreporting of COIs occurred in 52% of the included CPGs and 8% of all CPG developers. Underreporting COI disclosures was negatively associated with low-quality CPG (Odds ratio [OR], 0.57; 95% confidence interval [CI]: 0.11, 3.04). On the other hand, CPGs that had more than 13% of CPG developers with voting rights on recommendation decisions and underreporting of COI disclosures were positively associated with low quality (OR, 1.78; 95% CI: 0.25, 12.45). For individual CPG developers with voting rights for recommendation decisions, the presence of a COI was positively associated with low quality (OR, 1.11; 95% CI: 0.71, 1.75). This study demonstrates that the involvement and underreporting of COIs did not seriously distort the CPG development process. However, the COI-related factors of CPG developers with voting rights for recommendation decisions may be associated with low CPG quality.

## 1. Introduction

Understanding the influence of conflicts of interest (COI) on scientific evidence is important. A Cochrane methodological review reported an association between industry funding and favorable conclusions in original primary studies, which were mainly clinical trials [1]. Consistent results on the bias associated with financial COIs have been reported in systematic reviews and clinical practice guidelines (CPGs) [2,3]. Particularly, clinical practice guidelines have a significant impact on the clinical actions and prescribing behaviors of physicians; thus, having transparency in the relationship between CPGs and conflicts of interest is crucial. CPG development is divided into three processes: literature search, voting, and recommendation-making, and COIs can potentially affect these three processes.

The current international standard COI policy in CPG development requires full COI transparency to eliminate financial influence on guideline development [4,5]. However, in reality, the amount that needs to be disclosed varies among different guidelines. In 36 CPGs published in 2018–2019, only two (6%) organizations set a COI disclosure threshold higher than $0 [6]. Particularly in Japan, many CPGs require disclosure only when the amount exceeds a certain threshold. On the other hand, almost all CPGs from Japanese professional medical associations set a threshold for COI disclosures higher than $4587 (JPY 1 million). The eligibility criteria for CPG developers include the income of an officer or advisor of a company or for-profit organization (one million yen or more per company/year) [7]. It is believed that the current situation, where the standards for financial disclosure in CPGs vary by country, needs improvement.

In addition, previous studies examining the effect of COIs on CPGs have revealed discrepancies in COI disclosures [8,9]. In 2019, a systematic review reported that 32% of the authors had undisclosed financial conflicts [10]. In Japan, analyses of COI disclosures in each CPG revealed that many CPG developers did not disclose COIs that exceeded the declaration threshold [11,12], indicating the presence of undisclosed and underreported COIs related to CPGs. Therefore, the proportion of COIs in CPG documents based on self-disclosure is sometimes underestimated compared to the actual proportion of COIs.

The rigor of guideline development and editorial independence have the strongest influence on the overall guideline quality and recommendations for use [13]. Thus, it is important to consider the impact of underreporting COI disclosures on the quality of CPGs, including the rigor of development, because underreporting hinders transparency and raises concerns regarding the potential impact of these COIs. In Japan, 30 medical societies primarily take responsibility for specialist medical education. These societies are involved not only in specialist medical training and physician education but also in creating CPGs that form the basis of medical practice in their respective fields. Despite playing such a crucial role, it has been shown that the directors of these major societies often have strong financial conflicts of interest with pharmaceutical companies [14]. In this context, examining the COI status of authors involved in CPGs, their disclosure status, and the impact of undeclared COIs is important to clarify the importance of COI transparency in the preparation of CPGs. To the best of our knowledge, studies evaluating the influence of underreporting COI disclosures on the quality of CPGs are scarce. This study aimed to examine the proportion of underreported COI disclosures in Japanese CPGs. Additionally, we estimated the association between COI-related factors and CPG quality using the Appraisal of Guidelines for Research and Evaluation II (AGREE II).

## 2. Materials and Methods

### 2.1. Study Design and Protocol

This was a cross-sectional meta-epidemiological study. We followed the meta-epidemiological study reporting guidelines [4] (Appendix A). We published this protocol in osf.io (https://osf.io/p35kt/, accessed on 17 November 2023). We partially used our unpublished data for secondary analysis of a previous study [15].

### 2.2. Study Selection

A previous study included 53 systematic review (SR)-based CPGs published in 2018 and 2019 by 30 Japanese professional medical associations as the basis for training specialties in the field [15]. We defined SR-CPGs as “statements that include recommendations intended to optimize patient care” and an assessment of the benefits and harms of alternative care options based on the Institute of Medicine (IOM) criteria [5]. We selected 23 SR-based CPGs published in 2019 from a previous study [15], and then we focused more on recent CPGs (Figure 1, Appendix A).

### 2.3. Data Extraction and Assessment

#### 2.3.1. Exposures

We selected the financial COIs of drug companies as measurable data. The unit of analysis in this study was defined as each CPG or an individual CPG developer. For the latter, we specifically focused on CPG developers who had the right to vote on recommendation decisions for the clinical question (CQ) since they are involved in the quality of the COI.

The main exposure in this study was the underreporting of COI disclosures based on each COI policy in eligible CPGs in Japan. The unit of analysis was the CPG. The underreporting of a COI disclosure was defined as an underreported financial COI, considering the discrepancy between the CPG developer’s self-disclosure and publicly disclosed payment data by drug companies. When at least one CPG developer was involved in the underreporting of a COI disclosure, the CPG was categorized as an underreported COI disclosure.

The secondary exposure was the presence of a financial COI without regard to the threshold for COI disclosure according to the global standard policy [6], and the CPGs with more than 5% or 13% of the CPG developers underreporting COI disclosures, who had the right to vote on the recommendation decision. Based on the median and third quartile of the data distribution, we set 5% or 13% as the cut-off points. The additional exposure was the individual CPG developers, who had the right to vote on recommendation decisions for CQs. The unit of analysis was the CPG developer.

We confirmed the COI statement of the CPG developer and the threshold for COI disclosures prescribed by each CPG in the CPG document. Most CPGs followed the threshold (payments of 500,000 Japanese yen or more [equal to approximately 4587 USD]/company/year/item for speaking, writing, or lecturing and payments of 1,000,000 yen or more [equal to approximately 9174 USD]/company/year/item for consulting, research, or scholarship donations) for each COI disclosure [7]. When the statement was insufficient, we contacted the Japanese professional medical associations or guideline development committees responsible for publishing these guidelines to confirm the details of the COI statements and policies.

We collected information on the payments for speaking, writing, and consulting (more than 4587 USD) from a publicly accessible payment database maintained by the Medical Governance Research Institute [16]. This database collects payment data for speaking, writing, and consulting purposes disclosed by 92 pharmaceutical companies affiliated with the Japan Pharmaceutical Manufacturers Association (JPMA) [17]. The JPMA is the largest Japanese pharmaceutical trade association. Since 2013, the JPMA has required all member companies to disclose payments to medical institutions and healthcare providers on their web pages [18]. Regarding this period, we collected payment data from 2016 to 2018 because most Japanese professional medical associations ask for COI declarations three years before the publication of a guideline. Additionally, we collected data two years after a guideline publication because some organizations, including the Institute of Medicine, recommend that guideline developers remain free of conflicts for this time period after the guideline is published [19].

#### 2.3.2. Outcomes

The main outcome was the domain 3 (rigor of development) score using the Japanese version of AGREE II in each SR-CPG [20], which is a standard tool used to appraise the methodological quality of CPGs. This tool consists of 23 items that evaluate six domains.

In a previous study, four authors independently evaluated the AGREE II scores of eligible CPGs [15]. We defined high-quality CPGs as those scoring >60% in domain 3 and low-quality CPGs in the remaining cases, based on the standard criteria [21].

#### 2.3.3. Variables

We used data on the characteristics of the individual CPGs from a previous study [15]. The CPG characteristics gathered were as follows: names of the committed Japanese professional medical associations, funding (government, Japanese professional medical associations), number of guideline developers, number of CQs or recommendations, number of panelists, number of SR team members, involvement of CPG methodologists, involvement of patients in the panel, right to vote on the recommendation’s decision in the CQ, adopted guideline development methods, free accessibility to CPGs, and accessibility to SR.

The characteristics of the individual CPG developers were as follows: gender, physician, affiliations (university or university hospitals, general hospitals, research institutes, and clinics), and university professors. Both reviewers independently confirmed these characteristics, and disagreements were resolved through discussions. If variables and processes in the CPGs were uncertain, the Guideline Development Committee was contacted for clarification.

#### 2.3.4. Statistical Analysis

Descriptive statistics were used to summarize the results. The Japanese yen (¥) was converted into the corresponding USD ($) using the 2019 average monthly exchange rate of 109.0 JPY per 1 USD. We performed a univariate logistic regression analysis to evaluate the association between COI and CPG quality and calculated the odds ratio (OR) and 95% confidence interval (CI). The statistical significance level was set at 5%. We used Stata, ver. 17.0 (StataCorp LLC, College Station, TX, USA).

## 3. Results

The COI statements and financial COIs from drug companies (2016–2018) for developers of the 23 CPGs are summarized in Table 1. The threshold for disclosing COIs in the CPG documentation was clearly described by 43% (10/23) of the CPGs. After our inquiries, the remaining 13 CPGs stated a threshold for COI disclosure. A total of 52% (12/23) of the CPGs included CPG developers who underreported a COI disclosure. The number of CPG developers underreporting COI disclosures was 8% (94/1114). Among CPG developers with the right to vote on the recommendation decision in the CQ, the underreporting of COI disclosures was 10% (40/364).

CPG developers with financial COIs from drug companies accounted for 20% (223/1114). The median financial COI in the three years (2016–2018) was 36,093 USD (Interquartile range [IQR], 12552 USD–87010 USD). From 2016 to 2020, there were more COIs for speaking than for writing and consulting (Appendix A). Of the 223 CPG developers with COIs from 2016 to 2018, 164 (73%) had COIs in 2019 and 2020.

The characteristics of the 23 CPGs were separately reported as low or high AGREE II scores (Table 2). The proportion of adopted guideline development methods, such as the Grading of Recommendations, Assessment, Development, and Evaluation approach (GRADE) and Minds, was similar between the two groups. The main characteristics of the CPGs with low AGREE II scores were as follows: government funds (8%), no involvement of CPGs methodologists (15%), and patients in the panel (15%). The main characteristics of individual CPG developers with low AGREE II scores were as follows: female (5%) and general hospitals (7%).

Table 3 shows the univariate associations between CPGs with low AGREE II scores and COI-related factors. For CPGs, the underreporting of COI disclosures was negatively associated with low quality (OR: 0.57; 95% CI: 0.11, 3.04). Similarly, the presence of a COI was negatively associated with a low quality of life (OR: 0.57; 95% CI: 0.11, 3.04). More than 13% of CPG developers with underreporting of COI disclosures who had the right to vote on the recommendation decision were positively associated with low quality (OR: 1.78; 95% CI: 0.25, 12.45), although this was not significant. For individual CPG developers who had the right to vote on the recommendation’s decision, the presence of a COI was positively associated with low quality (OR: 1.11; 95% CI: 0.71, 1.75); however, this was not significant.

## 4. Discussion

The present study is a secondary analysis of a previous study that evaluated the methodological quality of Japanese CPGs published in 2019 and COI-related factors for 23 CPGs and 1114 CPG developers. In 52% of the included CPGs, CPG developers underreported their COI disclosures. CPG developers with underreported COI disclosures account for 8% of the total. Underreporting of COI disclosures is negatively associated with low-quality CPG. On the other hand, more than 13% of CPG developers underreporting COI disclosures who had the right to vote on the recommendation decision were positively associated with low quality. Therefore, our findings highlight the importance of carefully assessing the potential impact of COIs on CPG development, particularly in the context of recommendation decisions.

Japanese CPG developers do not disclose their financial COIs accurately. Despite the high threshold for COI disclosure in Japanese CPGs, as many as 8% of CPG developers have underreported COI disclosures. Our results align with those from previous research studies on COI disclosures within the realm of CPGs. A recent systematic review of financial COIs in CPG reported that 45% of nearly 15,000 guideline authors had at least one financial COI; however, many of them were undisclosed [10]. The undisclosed financial COIs in psoriatic arthritis CPG developers in Japan and the USA were reported based on major Japanese pharmaceutical companies and the USA Open Payments database from 2016 to 2018 [22]. A total of 18 authors (78.3%) in Japan and 12 authors (48.0%) in the USA had undisclosed financial COIs worth $474,663 and $218,501, respectively. In the 2012–2014 Australian CPGs, 24% of CPG developers had at least one potentially relevant undisclosed COI [19]. Of these undisclosed relationships, the first category of relationship listed in the relevant disclosure was pharmaceutical company grants (64%) or personal fees (36%). When the time frame was extended to 3 years after the guideline, the proportion of potentially relevant undisclosed relationships increased from 24% to 28%. Our findings are consistent with the fact that COI disclosure policies in CPGs are violated and that this issue is a global concern.

COI-related factors as a unit of CPG, such as underreporting COI disclosures or the presence of a financial COI, may be negatively associated with low-quality CPGs. However, a COI generally results in a low CPG quality [5,15]. Our results indicate that the use of guideline development methods (GRADE and Minds) has a stronger effect on domain 3 of AGREE II than on COI-related factors. As for other influencing factors, the guideline developers/clinical question ratio is related to the high rigor of development quality of AGREE II [15]. However, we could not clarify this association using multivariable regression analysis because of the small sample size.

The COI-related factors of CPG developers, who have the right to vote on recommendation decisions, might be associated with low-quality CPGs. Although their expertise is valuable, COI factors can potentially compromise the objectivity and quality of guidelines [15]. A recent meta-epidemiological study reported that consensus-based guidelines produce more recommendations that violate the principles of evidence-based medicine than evidence-based guidelines [23]. CPGs produced by developers with COIs are more likely to contain biases in their recommendations, raising concerns about the integrity and transparency of the development process [24]. Thus, to avoid bias by COI factors, guideline committees must be more careful about the COI of CPG developers with voting rights. The Guidelines International Network recommends that no one with relevant COIs should decide the direction or strength of a recommendation [25].

Our results suggest actions to address COI-related problems in Japanese CPGs. First, appropriate COI disclosures are required for Japanese CPG developers. A more rigorous cross-checking system is required for COI information, which is self-reported by CPG developers. One feasible strategy is to use a publicly available COI disclosure database [26]. Second, CPGs should clearly define financial COIs and their involvement in the recommendation-making process. As a general rule for guidelines in Japan, standardized descriptions would be meaningful. One option is to rate the COI according to the risk of influencing decisions in a specific guideline [27]. Third, to enhance the quality of CPGs, the guideline committee should select CPG developers with voting rights who do not have a COI involvement.

This study has some limitations. First, assessing only financial COIs (speaking, writing, and consulting) underestimates the prevalence and magnitude of COI disclosure violations. Financial disclosures in Japan include professional income, stock options, and research funding [7]. We also did not evaluate financial COIs from drug companies that are not members of the JPMA or from medical device manufacturers. Second, assessing only financial COIs worth more than 4587 USD underestimated the overall prevalence and magnitude of financial COIs. Third, the findings had limited external validity owing to the unique COI disclosure rules for CPG developers in Japan. Other regions may have varying COI rules and CPG development processes, thereby altering the outcomes. Therefore, considerable discretion must be exercised when applying these results to other settings.

## 5. Conclusions

The involvement and underreporting of COIs did not seriously distort the CPG development process. However, the COI-related factors of CPG developers, who have the right to vote on recommendation decisions, might be associated with low CPG quality. Guideline committees in Japan should consider the influence of COIs for a transparent and rigorous guideline development process to ensure high-quality standards for patient care.

## Figures and Tables

**Figure 1 jpm-13-01722-f001:**
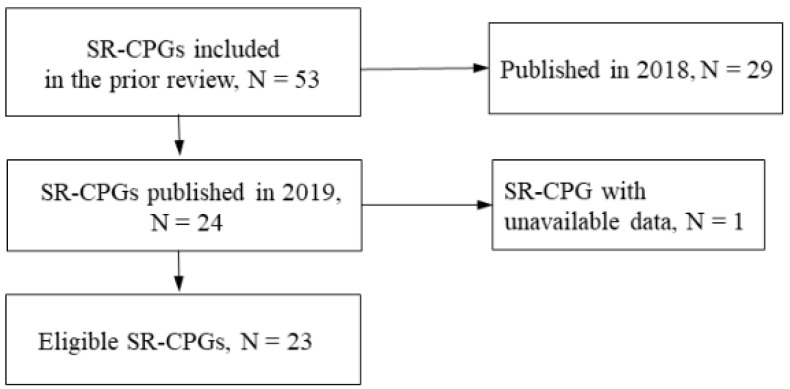
Flowchart for the selection of the eligible SR-CPGs. SR-CPGs: Systematic review-based clinical practice guidelines. A previous review indicates this citation [15].

**Table 1 jpm-13-01722-t001:** Financial COIs from drug companies (2016 to 2018) for the CPG developers in 23 Japanese CPGs published in 2019.

	*n* (%)
CPGs (*n* = 23)	
CPGs described the threshold for COI disclosure in CPG documentation	10 (43)
CPGs reported the threshold for COI disclosure after inquiry	13 (57)
CPGs including CPG developers with underreporting COI disclosure	12 (52)
CPG developers (*n* = 1114)	
CPG developers with underreporting COI disclosure	94 (8)
CPG developers with voting rights on recommendation decision in CQ, with underreporting COI disclosure	40 (4)
CPG developers with the presence of COI	223 (20)
CPG developers with payments > $5000	205 (18)
CPG developers with payments > $10,000	177 (15)
CPG developers with payments > $100,000	51 (4)

CPG, clinical practice guideline; COI, conflict of interest; CQ, clinical question.

**Table 2 jpm-13-01722-t002:** Characteristics of the included 23 Japanese clinical practice guidelines published in 2019.

		CPGs with Low AGREE II Score (*n* = 13)	CPGs with High AGREE II Score (*n* = 10)	Total (*n* = 23)
The characteristics of individual CPGs				
Fund				
	Government	1 (8%)	3 (30%)	4 (17%)
	Japanese professional medical associations	10 (77%)	7 (70%)	18 (74%)
	Unclear	2 (15%)	0 (0%)	2 (9%)
Number of guideline developers		24 (14–48)	47 (34–59)	35 (18–56)
Number of CQs or recommendations		21 (16–28)	10.5 (8–20)	16 (11–28)
Number of panelists		9 (0–13)	11 (6–16)	9 (0–14)
	Unclear	4 (31%)	2 (20%)	6 (26%)
Number of SR team members		10 (0–31)	21 (4–28)	13 (0–31)
	Unclear	6 (46%)	2 (20%)	8 (35%)
Involvement of CPGs methodologists		2 (15%)	8 (80%)	10 (43%)
Involvements of patients in the panel		2 (15%)	4 (40%)	6 (26%)
Adopted guideline development methods				
	GRADE	2 (15%)	2 (20%)	4 (17%)
	Minds 2014 or after	9 (70%)	7 (70%)	16 (70%)
	Minds 2007	1 (8%)	0 (0%)	1 (4%)
	Unclear	1 (8%)	1 (10%)	2 (9%)
Freely accessible to CPGs		10 (77%)	9 (90%)	19 (83%)
Accessible to SR		5 (39%)	6 (60%)	11 (49%)
The characteristics of individual CPG developers				
Number of individual CPG developers		560	554	
Gender				
	Male	533 (95%)	399 (72%)	
	Female	27 (5%)	77 (14%)	
Physician		554 (99%)	482 (87%)	
Affiliation				
	University or university hospital	346 (62%)	396 (71%)	
	General hospital	38 (7%)	297 (54%)	
	Research institute	1 (0.2%)	12 (2%)	
	Clinic	8 (1%)	3 (1%)	
	Others	4 (1%)	9 (2%)	
University professor		136 (24%)	237 (43%)	

Data are presented as number (%) or median (interquartile range). CPGs, clinical practice guidelines; AGREE II, Appraisal of Guidelines for Research & Evaluation II; CQs, clinical questions; GRADE, grading of recommendations assessment, development and evaluation approach; SR, systematic review.

**Table 3 jpm-13-01722-t003:** Association between CPGs with low AGREE II score and COI.

	OR (95%CI)
All CPGs (*n* = 23)	
Underreporting COI disclosure	0.57 (0.11 to 3.04)
CPG developers with underreporting COI disclosure, who have voting rights on the recommendation decision	
More than 5% More than 13%	0.57 (0.11 to 3.04) 1.78 (0.25 to 12.45)
The presence of COI	0.37 (0.06 to 2.09)
All CPG developers who have voting rights on the recommendation’s decision (*n* = 364)	
Underreporting COI disclosure	0.87 (0.45 to 1.68)
The presence of COI	1.11 (0.71 to 1.75)

CPG, clinical practice guideline; COI, conflict of interest; OR, odds ratio; CI, confidence interval.

## Data Availability

Data supporting the findings of this study are available from the corresponding author upon request.

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
