# Peer review of "Association between Conflicts of Interest Disclosure and Quality of Clinical Practice Guidelines in Japan: A Meta-Epidemiological Study"

_jpm, 2023, doi:10.3390/jpm13121722_

Round 1
Reviewer 1 Report
Comments and Suggestions for Authors
The meta-epidemiological study examining the association between conflicts of interest disclosure and the quality of clinical practice guidelines in Japan have potential impact on healthcare practices. This study's findings provide invaluable insights into the correlation between transparency in disclosures and the quality of guidelines, offering a significant contribution to the field. However, some things in this study needs to be clarified:
1. Can you please explain why in Methodology section the cut off CPGs with more than 5% or 13% of the CPG developers that underreport COI disclosure was established?
2. How do you explain that underreporting of COI disclosures is negatively associated with low-quality CPG?
3. Can you provide more data in discussion part regarding other countries and their COI-related problems and solutions?
Author Response
Reviewer 1
The meta-epidemiological study examining the association between conflicts of interest disclosure and the quality of clinical practice guidelines in Japan have potential impact on healthcare practices. This study's findings provide invaluable insights into the correlation between transparency in disclosures and the quality of guidelines, offering a significant contribution to the field. However, some things in this study needs to be clarified:
Response: We thank the Reviewer 1 very much for these comments. We have revised the related text according to the comments.
- Can you please explain why in Methodology section the cut off CPGs with more than 5% or 13% of the CPG developers that underreport COI disclosure was established?
Response: The median (interquartile range) was 5% (0-13%). We determined that the first quartile (0%) was not useful for the cut-off point. Then, we selected 5% and 13% as the cut-off points.
Line 112: Based on the median and third quartile of the data distribution, we set 5% or 13% as the cut-off points.
- How do you explain that underreporting of COI disclosures is negatively associated with low-quality CPG?
Response: As the Reviewer 1 pointed out, the results are not consistent with the previous studies. We discussed it and edited the following sentences.
Line 234: Our results indicate that the use of guideline development methods (GRADE and Minds) has a stronger effect on domain 3 of AGREE II than on COI-related factors. As for other influencing factors, the number of guideline developers/clinical question ratio is related to the high rigor of development quality of AGREE II [1]. However, we could not clarify this association using multivariable regression analysis because of the small sample size.
[1] Kataoka Y, Anan K, Taito S, Tsujimoto Y, Kurata Y, Wada Y, Maruta M, Kanaoka K, Oide S, Takahashi S, Nango E. Quality of clinical practice guidelines in Japan remains low: A cross-sectional meta-epidemiological study. J Clin Epidemiol. 2021 Oct;138:22-31. doi: 10.1016/j.jclinepi.2021.06.025.
- Can you provide more data in discussion part regarding other countries and their COI-related problems and solutions?
Response: We agreed with the Reviewer 1’s suggestion. We provided other countries’ data (e.g., USA, Australia) in discussion. We have added the following sentences.
Line 230: The undisclosed financial COI in psoriatic arthritis CPG developers in Japan and USA was reported based on major Japanese pharmaceutical companies and USA Open Payments database from 2016 to 2018 [1]. A total of 18 authors (78.3%) in Japan and 12 authors (48.0%) in the USA had undisclosed financial COI worth $474,663 and $218,501, respectively. In 2012–2014 Australian CPGs, 24% of CPG developers had at least one potentially relevant undisclosed COI [2]. Of these undisclosed relationships, the first category of relationship listed in the relevant disclosure was pharmaceutical company grants (64%) or personal fees (36%). When the time frame was extended to 3 years after the guideline, the proportion of potentially relevant undisclosed relationships increased from 24% to 28%.
Line 245:CPGs produced by developers with COI are more likely to contain biases in their recommendations, raising concerns about the integrity and transparency in the development process [3]
Line 257: The Guidelines International Network recommends that no one with relevant COIs should decide the direction or strength of a recommendation [4].
[1] Mamada H, Murayama A, Kamamoto S, Kaneda Y, Yoshida M, Sugiura S, Yamashita E, Kusumi E, Saito H, Sawano T, Tanimoto T, Vassar M, Ozieranski P, Ozaki A. Evaluation of Financial and Nonfinancial Conflicts of Interest and Quality of Evidence Underlying Psoriatic Arthritis Clinical Practice Guidelines: Analysis of Personal Payments From Pharmaceutical Companies and Authors' Self-Citation Rate in Japan and the United States. Arthritis Care Res (Hoboken). 2023 Jun;75(6):1278-1286.
[2] Moynihan R, Lai A, Jarvis H, Duggan G, Goodrick S, Beller E, Bero L. Undisclosed financial ties between guideline writers and pharmaceutical companies: a cross-sectional study across 10 disease categories. BMJ Open. 2019 Feb 5;9(2):e025864. doi: 10.1136/bmjopen-2018-025864.
[3] Kim SY. Recent Advance in Clinical Practice Guideline Development Methodology. Korean J Fam Med. 2022 Nov;43(6):347-352. doi: 10.4082/kjfm.22.0178.
[4]Schünemann HJ, Al-Ansary LA, Forland F, et al. Guidelines International Network: Principles for Disclosure of Interests and Management of Conflicts in Guidelines. Ann Intern Med. 2015;163(7):548-553.
Reviewer 2 Report
Comments and Suggestions for Authors
This is a well written study. The objective is clearly stated and the conclusion is sufficiently supported by the findings. There is only one comment to improve the paper.
The CPGs were derived from a previously published systematic review. Please explain more clearly how 53 reviews were narrowed down to 23?
Author Response
Reviewer 2
This is a well written study. The objective is clearly stated and the conclusion is sufficiently supported by the findings. There is only one comment to improve the paper.
Response: We thank the Reviewer 2 very much for these comments.
The CPGs were derived from a previously published systematic review. Please explain more clearly how 53 reviews were narrowed down to 23?
Response: We thank the Reviewer 2 for this suggestion. A previous study included 53 systematic review (SR)-based CPGs published in 2018 and 2019. We chose SR-CPG published in 2019 because we were interested in more recent CPGs. As the figure 1 shows, one SR-CPG published in 2019 was excluded due to unavailable data. We searched the data in the documents and inquired with the CPG committee, but never heard back from them. We have edited the following sentences.
Line 85: A previous study included 53 systematic review (SR)-based CPGs published in 2018 and 2019 by 30 Japanese professional medical associations as the basis for training specialties.